# Unmet Need for Oral Corticosteroids Use and Exacerbations of Asthma in Primary Care in Taiwan

**DOI:** 10.3390/biomedicines10123253

**Published:** 2022-12-14

**Authors:** Yen-Wen Chen, Yi-Han Hsiao, Hsin-Kuo Ko, Tien-Hsin Jeng, Kang-Cheng Su, Diahn-Warng Perng

**Affiliations:** 1Departments of Chest Medicine, Taipei Veterans General Hospital, Taipei 11217, Taiwan; 2School of Medicine, National Yang Ming Chiao Tung University, Taipei 30010, Taiwan; 3Institute of Clinical Medicine, National Yang Ming Chiao Tung University, Taipei 30010, Taiwan

**Keywords:** exacerbation, primary healthcare facility, oral corticosteroid, severe asthma, short-acting β2-agonist

## Abstract

Patients with asthma are treated in primary healthcare facilities (PHCFs) in Taiwan, where the asthma control status associated with acute exacerbation (AE) and use of oral corticosteroids (OCS) and short-acting β2-agonist (SABA) inhalers remains unclear. A cross-sectional, close-ended, face-to-face questionnaire survey invited board-certified physicians who treat adult asthma patients in PHCFs. The contents of the questionnaire included three parts: rescue OCS to treat AE, regular OCS for asthma control, and AE-related adverse outcomes. There were 445 out of 500 physicians who completed the questionnaire, with 61% of them being non-pulmonologists. A substantial proportion of asthma patients needed rescue OCS or regular OCS each month, or ≥3 canisters of SABA inhalers per year. Approximately 86% of physicians reported their patients with ≥2 AE-related unscheduled visits to clinics or emergency departments in the past year. A total of 41% of physicians reported their patients receiving intubation or intensive care in the past year. A total of 92% of physicians prescribed rescue OCS ≤ 40 mg/day. A total of 92% of physicians prescribed rescue OCS for a duration of ≤7 days for AEs. A total of 85% of physicians prescribed regular OCS ≤ 10 mg/day for asthma control. This is the first study to present the perceptions of asthma-treating physicians on the use of OCS in PHCFs. In summary, 31% of physicians reported ≥ 6% of their patients needed OCS for asthma control and 41% of physicians reported their patients with adverse outcomes in the past year. Thus, the need to improve asthma control in Taiwan is suggested by our study results.

## 1. Introduction 

Asthma is one of the most serious health concerns worldwide. According to the World Health Organization, asthma affected an estimated population of 262 million people in 2019 and caused 461000 deaths [1]. Airway inflammation is a key component of asthma, and anti-inflammatory treatment is a crucial therapeutic strategy for asthma control. Inadequate treatment of asthma may lead to poor control of symptoms and acute exacerbation (AE) which may require unscheduled clinic visits, emergency department (ED) visits, and hospitalizations. The exacerbations will also lead to increases in total healthcare costs and subsequent socioeconomic burdens [2].

Oral corticosteroids (OCS) are considered the most effective short-term treatment for AE of asthma and long-term management for uncontrolled severe asthma (SA) despite their significant side effects [3]. Recently, the Optimum Patient Care Research Database in the UK reported that the prevalence of OCS-related comorbidities in SA patients ranged from 4% to 65%, including dyspeptic disorders (65%), obesity (42%), hypertension (34%), osteoporosis (16%), type II diabetes (10%), and sleep disorder (4%) [4]. In a 10-year analysis from the National Health Insurance Research Database (NHIRD) in Taiwan, OCS remained the second most commonly used anti-asthmatic medication and ranged from 21.7% to 35.8% between primary care clinics and different medical institutions [5]. Both short- and long-term use of OCS increased the risk of corticosteroid-related complications [6,7]. In addition, the increased use of short-acting β2-agonist (SABA) and dosage of OCS in outpatient clinics was reported to be associated with the increased risk of asthma death [8].

According to a survey that was conducted in the UK to investigate the asthma population in primary care practices, frequent exacerbators, are defined as those requiring ≥ 2 rescue OCS courses for AEs in the past 12 months, ranging from 4% to 14% [9]. According to the Global Initiative for Asthma (GINA) guidelines [3], asthma patients who have ≥2 AEs per year or uncontrolled symptoms despite medium- or high-dose inhaled corticosteroids (ICS) should be referred to respiratory specialists. In SA, many biological agents can achieve symptom control, reductions in AE rates, and maintenance of OCS dosing in large placebo-controlled trials [10,11]. An accurate diagnosis, comprehensive assessment of disease severity, understanding of disease mechanisms, avoiding reliance on OCS and SABA, and adequate selection of cases for treatment with biological agents are needed to effectively treat uncontrolled asthma [12].

In Taiwan, more than 80% of asthmatics were treated by physicians in primary healthcare facilities (PHCFs), while 12% were treated by pulmonologists [5]. Therefore, this study aimed to investigate the perspectives of physicians in PHCFs using a face-to-face questionnaire survey focusing on the current status of OCS use, SABA reliance, and frequencies of AE and AE-related adverse outcomes in the management of asthma patients in Taiwan.

## 2. Materials and Methods

### 2.1. Study Design and Participants

The study was designed as a cross-sectional, close-ended questionnaire survey (Appendix A) to collect insights from physicians at PHCFs in Taiwan. The participants were eligible if they were registered and board-certified physicians and were experienced in treating asthma patients aged greater than 18 years. Participants could possess different medical specialties and practice in doctors’ offices or community hospitals. The participating physicians were sampled from the registered healthcare institution lists, invited by phone calls, and followed up by face-to-face interviews using a pre-specified questionnaire. This study was approved by the Institutional Review Board of the Taipei Veterans General Hospital (ID: 2021-05-001BC). Raw data were generated at the Taipei Veterans General Hospital large-scale facility. Derived data supporting the findings of this study are available from the corresponding author upon request.

### 2.2. Questionnaire

A simple, short questionnaire was developed by the study team based on a literature review, consisting of appropriate items relevant to the current practices involving asthma AEs and the use of OCS. The questionnaire comprises 15 questions and is categorized into two parts—the backgrounds of the physicians and the physicians’ perspectives on asthma management. The entire questionnaire is presented in the Appendix A.

### 2.3. Date Analysis

Data were analyzed using the statistical software SPSS (version 20.0, IBM Corp., Armonk, NY, USA). Associations between categorical data, including clinical settings, medical specialty, and proportion of responses to questions were analyzed using the chi-squared test. Differences were considered significant at *p* value < 0.05.

## 3. Results

### 3.1. Background of Asthma Treating Physicians

This study was conducted between 20 May and 31 August 2021. A total of 445 participants in PHCFs were invited, and all completed the interviews. The responding documents were valid, without any significant missing data. In this survey, the leading medical specialty was pulmonology (39.1%), followed by family medicine (18%) and pediatrics (17.8%). The medical practice was divided into doctors’ offices (64.3%)- and community hospitals (35.7%) (Table 1). Most pulmonologists (83.9%) practiced in community hospitals while most non-pulmonologists (95.2%) were in doctors’ offices. The physicians’ treatment experience, based on their average monthly asthma patient numbers, is ranked as follows: <50 patients (59.3% of total physicians), 51–100 (25.4%), and >100 (15.3%).

### 3.2. Oral Corticosteroid for Acute Exacerbation

Approximately 55.5% of physicians reported that >6% of asthma patients experienced AEs that required OCS treatment every month. Moreover, 8.8% of physicians reported that >20% of asthmatics required rescue OCS (Figure 1A). In addition, 91.9% of physicians reported that they prescribed ≤40 mg of prednisolone (or equivalent) (Figure 1B) and 92.2% prescribed an OCS rescue course of ≤7 days (Figure 1C).

### 3.3. Regular Oral Corticosteroid for Asthma Control

Approximately 90.5% of physicians prescribed regular OCS for asthma control in ≤10% of asthmatics per month (Figure 2A). Approximately 85.4% of physicians prescribed ≤10 mg prednisolone equivalent (Figure 2B) and 64% prescribed a treatment course ≤1 month as regular OCS therapy (Figure 2C). In addition, 79.8% of physicians reported ≤10% of their asthmatics requiring ≥2 courses of regular OCS per year (Figure 2D). The two leading reasons for regular OCS use were the disease entity of asthma and non-adherence or misuse of inhalers, both of which were significantly rated among pulmonologists and non-pulmonologists (Table 2).

### 3.4. Asthma Control Status

While considering the percentage of asthmatics experiencing ≥2 unscheduled visits to the clinic or ED due to AEs in the past year, physicians reported that only 13.7% of asthmatics were event-free (Figure 3A). Of the total percentage of intubations or ICU admissions in the past year, the event-free proportion was 59.1% (Figure 3B). Asthma patients who used ≥3 canisters of SABA during a 1-year period were considered SABA over-users [13]. Nearly 50% of physicians reported that 6% to 50% of asthmatics overused SABA in the past year (Figure 3C). Up to 92.2% of physicians reported that 1% to 10% of asthmatics benefited from biological agents (Figure 3D).

### 3.5. Differential Concepts between Pulmonologists and Non-Pulmonologists

From the physicians’ perspectives based on medical specialty, pulmonologists rated their asthmatics with a higher proportion of rescue and regular OCS use, more unscheduled visits, more ICU admissions, a higher proportion of SABA over-users, and higher confidence in biologics-related benefits, when compared to non-pulmonologists (Table 3).

## 4. Discussion

In this study, we discovered that the majority of asthma-treating physicians (60.9%) in PHCFs were non-pulmonologists. From the physicians’ perspectives, there was a substantial proportion of asthmatics who require rescue OCS or regular OCS per month, or overuse SABA inhalers per year and experience poor asthma outcomes based on AE-related unscheduled visits to the ED, intubations, or ICU admissions. For the first time, we presented the perceptions of asthma-treating physicians in PHCFs, and these results indicate the need to improve asthma control in Taiwan.

In our previous study, based on the NHIRD between 2000 and 2005 [8] asthma mortality was significantly associated with the increased use of SABA inhalers and OCS, more frequent ED visits, and hospitalizations. Moreover, 25% of fatal asthma cases were prescribed three to five canisters of SABA inhalers, and up to 31% were prescribed ≥ 6 canisters of SABA in the previous year [8]. Recently, the SABRINA study based on the Swedish asthma registry revealed that one-third of asthma patients collected ≥3 canisters of SABA annually [13]. In the present study, nearly 50% of physicians rated 5% to 50% of asthmatics as overusing SABA inhalers. Overuse of SABA instead of regular use of controllers has been associated with increased risks of disease exacerbation and mortality [13]. Over the past 20 years, despite the update of asthma treatment guidelines, the persistent high proportion of SABA overuse indicates that this issue remains underestimated in Taiwan and needs urgent action to reduce the potentially fatal outcomes in the future.

In this study, up to 84.6% of the physicians reported that their asthmatics had undergone ≥2 AE-related unscheduled visits to the clinic or ED, as well as 40.9% reported their asthmatics had experienced ≥1 intubation or ICU admission in the past year. As per the National Health Interview Survey in the USA conducted from 2011 to 2016, 44.7% of adult asthmatics experienced ≥1 AE and 9.9% had ≥1 AE-related ED visit in the past year [14]. In the analysis based on the NHIRD in Taiwan, the rate of hospitalizations or ED visits in adult asthmatics decreased from 1.42% to 0.59% from 2000 to 2010 [5], and the ICU admission rate in hospitalized asthmatics was 5.4% during a 5-year (2001 to 2005) period [8]. In other studies that reported asthma AE-related hospitalizations, 10% were admitted to ICU in the USA [15] and Korea, [16] and 2% required intubation in the USA [15]. Furthermore, in an 8-year observational study in the USA, the utilization of invasive mechanical ventilation was 1.4% and 0.73%, as non-invasive ventilation was 0.34% and 1.9% in 2000 and 2008, respectively [17]. Taken together, the proportion of ED visits, intubations, and ICU admissions seems higher in this survey than in the previous NHIRD analysis in Taiwan and similar surveys conducted in other countries. This difference might be ascribed to different data sources (physicians’ vs. patient-based data). However, this study suggests that the adverse outcomes associated with asthma control in PHCFs may still be underestimated, particularly for those with SA.

In the early large-scale randomized controlled trial (FACET study) examining the effects of ICS with or without long-acting β2 agonists on asthma control in European countries, the rate of severe AE requiring OCS ranged from 19% to 39% in different treatment arms during the 1-year follow-up period [18,19]. Recently, in a population-based prospective cohort with a follow-up period of >10 years (Rotterdam study) in the Netherlands [20], the monthly exacerbation rate of asthmatics required from OCS 5.86% in summer to 10.51% in winter in real-world settings. Moreover, 70.3% of asthmatics experienced ≥1 AE per year and up to 6.2% had ≥10 AEs per year. In this study, only 44.5% of physicians considered the monthly AE rate of asthmatics requiring OCS to be ≤5%. In addition, over 90% of physicians from this survey prescribed oral prednisolone (or equivalent) ≤40 mg/day for ≤7 days as the treatment modality of rescue OCS for AE. These results are consistent with the international experts’ consensus [21], which suggests that 5–7 days is the usual maximal duration of a prednisolone dose of 0.5 mg/kg/day (or equivalents) for a short-course treatment for AE. Although the use of OCS rescue is still prevalent in different countries, the proportion of rescue OCS use in PHCFs in Taiwan is relatively high. Further investigations are needed to clarify the cause and to reduce OCS-related adverse outcomes.

As per current guidelines, regular OCS as part of maintenance treatment in asthma control is only recommended for those failing conventional treatment despite adequate inhaler use and management of comorbidities [3]. The proportion of regular OCS use largely varies in different countries, ranging from 20.7% in South Korea, 23.3% in the USA, 25% in Australia, and 59.6% in the UK [11]. The median daily dose of regular OCS in Australia was 10 mg/day (prednisolone equivalent), but with a wide dose range between 2 and 50 mg [22]. Recently, the international experts’ consensus proposed daily prednisolone ≤ 5 mg as an acceptable dose and considered a yearly cumulative dose of >0.5g or 1 g as an indicator of poor outcome [21]. Based on this consensus, given loose criteria of a yearly accumulative dose of 1g, this equals a daily use of 10 mg prednisolone for 3.3 months. In our study, 14.6% of physicians prescribed prednisolone equivalent > 10 mg/day, and 11.7% prescribed prednisolone for >3 months. Furthermore, 20.2% of physicians reported that >10% of their asthmatics required ≥2 courses of regular OCS per year. Clearly, a substantial proportion of asthma patients in Taiwan might be prescribed much higher doses of regular OCS that were beyond the proposed dose by the consensus. This finding might correspond to our previous 10-year NHIRD study, which revealed that a monthly prednisolone prescription of ≥110 mg in the past year for outpatients was an independent risk factor for fatal asthma [8]. Thus, we need to establish our own protocol for OCS tapering.

It is estimated that SA may affect 5–10% of asthma patients [23], and patients with SA may have increased hospitalizations, complications of OCS use, and poor health-related quality of life [24]. This asthma population may account for 50% of all asthma-related costs as reported by the World Allergy Organization [25]. We previously reported that 88% of asthma patients were treated by non-pulmonologists, and the OCS prescriptions were lower in medical centers (19.8%) compared to other clinical settings (26.3%), including doctors’ offices and community hospitals [5]. Based on these study results, we speculate that the participating physicians were treating a certain portion of SA patients, which might explain the relatively high proportion of OCS use, frequent ED visits, and ICU admissions, particularly for pulmonologists. With the emergence of biological agents, a subset of SA patients showed great improvement in asthma control [10]. However, most physicians (74.1%) in this survey rated that <5% of asthmatics may benefit from biological agents in the reduction of OCS doses and AE frequencies. In addition, pulmonologists expressed more confidence than non-pulmonologists on the effects of biologicals (Table 3) and were more likely to consider the disease entity of asthma as the major cause of regular OCS use, rather than non-adherence or misuse of inhalers. (Table 2). This observation implies pulmonologists might be more aware of the characteristics of SA and more willing to refer patients for treatment with biological agents. Currently, biological agents are a brand-new treatment modality for SA in Taiwan, and most SA patients requiring biological agents are treated in medical centers. Therefore, physicians in PHCFs, particularly non-pulmonologists, need more education programs to recognize SA patients and arrange timely referrals.

It is questionable whether pulmonologists reported worse asthma control status in this survey. Based on patient-level data, the characteristics of asthmatics treated by pulmonologists showcased higher asthma severity, more anti-asthmatic medication consumption, more comorbidities, and greater healthcare utilization than those treated by non-pulmonologists [26,27,28]. In contrast, this study showed the results based on physician-level observations. In addition, most pulmonologists practiced in community hospitals. Thus, it might be the speculation that pulmonologists are most likely to treat more complicated asthmatics in this study.

It may be argued that physicians in community hospitals were included and regarded as primary care physicians. Moreover, outpatients in hospitals in Taiwan are quite different from those in other countries because Taiwan has a unique healthcare system. Taiwan has a government-run, single-payer health insurance system, which is characterized by mandatory coverage for all citizens (almost >99%), convenient accessibility, low costs, and almost complete coverage of medical services. The system has neither a gatekeeper role nor restrictive referral regulations. That is, any outpatient has the freedom to choose any specialist/subspecialist in any hospital without a referral [29]. The Taiwan National Insurance Administration announced that a substantially high proportion of outpatients directly call hospital-based clinics without any referrals. Therefore, both doctors’ office- and hospital-based clinics are regarded as PHCFs in this study.

This study has some limitations. This study was based on a questionnaire survey that did not include patient-level data. Therefore, we could not exclude recall bias. Moreover, the different medical specialties and practice settings of the participating physicians might have some impact on the response results. The results might indirectly reflect the current status of asthma control in PHCFs, with a lack of precise data regarding important asthma control outcomes, such as SABA dispensing data, the rates of ED visits, hospitalizations, and ICU admissions.

## 5. Conclusions

A substantial proportion of asthmatics in PHCF are frequent exacerbators, SABA over-users, and potentially inappropriate OCS users. Both patients and physicians need to be educated on advanced asthma management. In particular, physicians in PHCFs need better recognition of SA patients and arrange timely referrals to optimize asthma control.

## Figures and Tables

**Figure 1 biomedicines-10-03253-f001:**
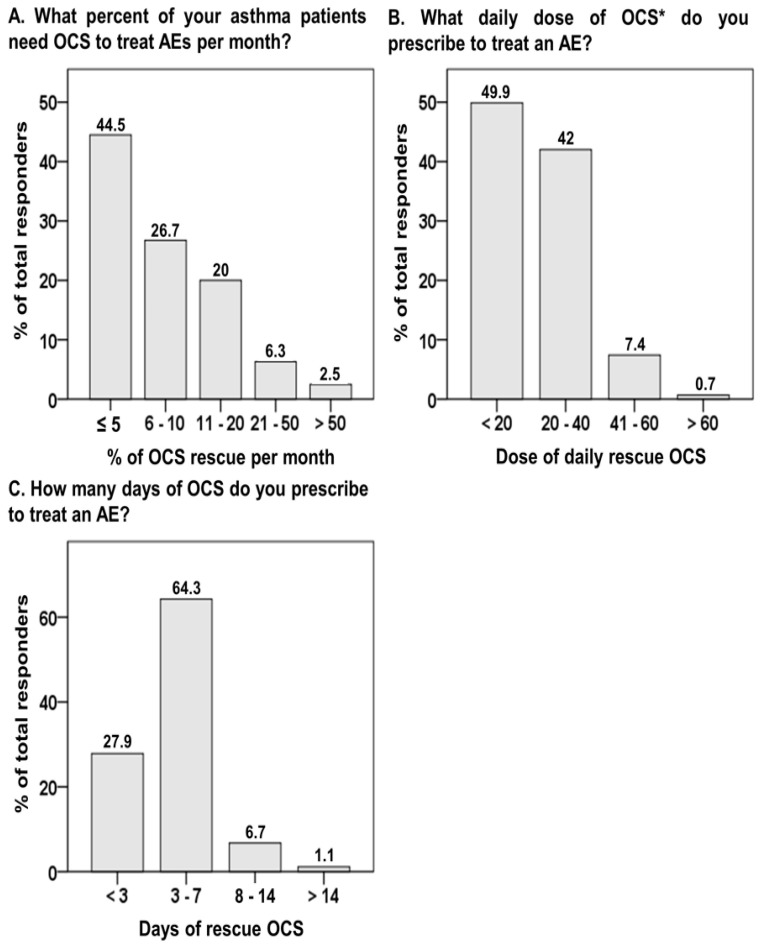
The proportion of responses to questions regarding oral corticosteroids for acute exacerbation of asthma. AE, acute exacerbation; OCS, oral corticosteroid. * indicates prednisone or equivalent in mg.

**Figure 2 biomedicines-10-03253-f002:**
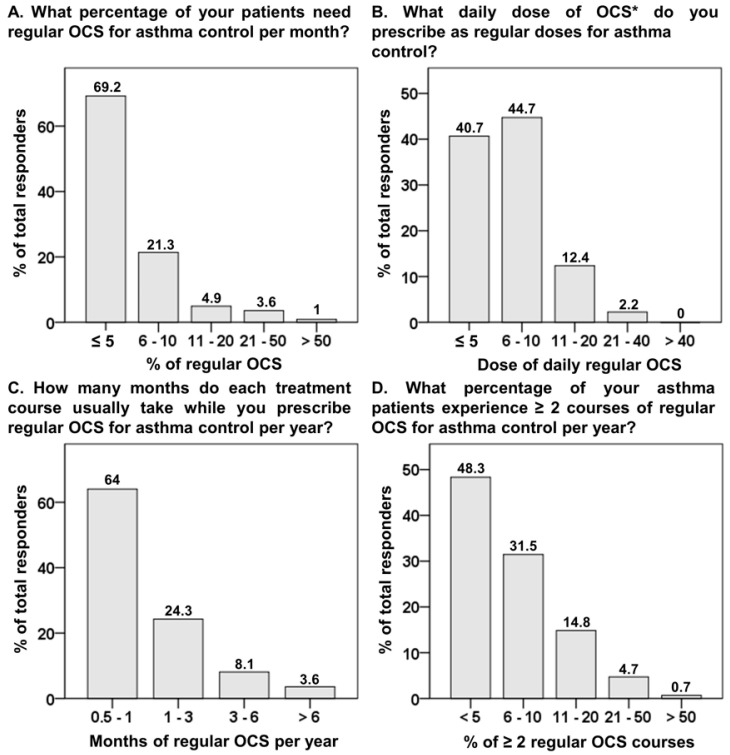
The proportion of responses to questions regarding regular oral corticosteroids for asthma control. OCS, oral corticosteroid. * Indicates prednisone or equivalent in mg.

**Figure 3 biomedicines-10-03253-f003:**
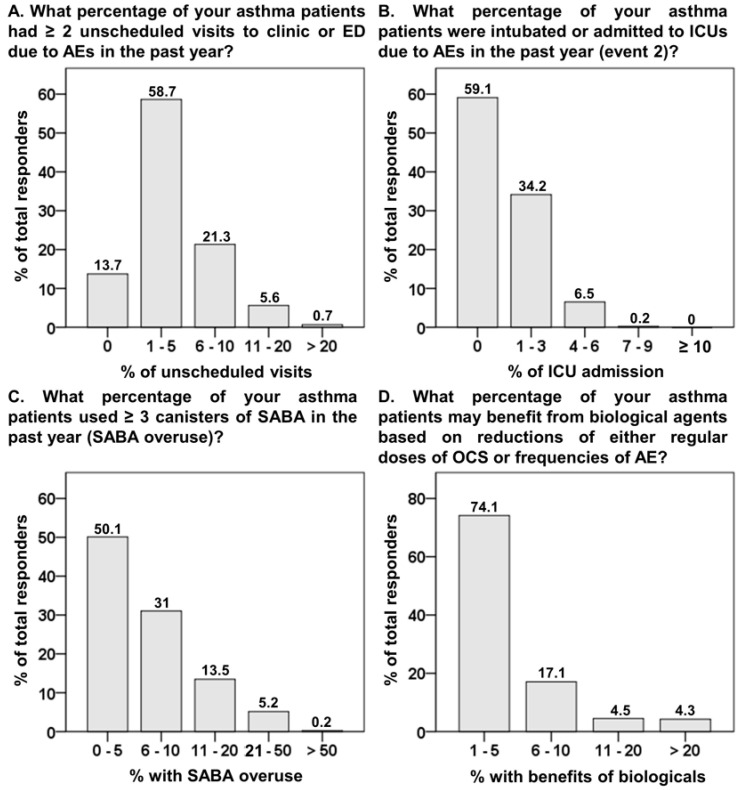
The proportion of responses to questions regarding asthma control status. AE, acute exacerbation; ED, emergent department; ICUs, intensive care units; OCS, oral corticosteroid; SABA, short-acting β2 agonists.

**Table 1 biomedicines-10-03253-t001:** The medical specialty and healthcare facility of participating physicians.

Healthcare Facility	Medical Specialty				
	Pediatrics (*n* = 79)	ENT (*n* = 60)	FM (*n* = 80)	IM (*n* = 52)	Pulmonology (*n* = 174)
Doctor’s office (*n* = 286)	78	59	73	48	28
Community hospital (*n* = 159)	1	1	7	4	146

ENT, Otorhinolaryngology; IM, internal medicine.

**Table 2 biomedicines-10-03253-t002:** Differential response regarding factors contributing to regular use of oral corticosteroids between pulmonologists and non-pulmonologists.

Do you Think Which Factors Contribute to Regular Use of OCS for Asthma Control? (Multiple Choices Allowed)	Total (*n* = 445)	Non-Pulmonologist (*n* = 271)	Pulmonologist (*n* = 174)
Disease entity of asthma	183 (41.1)	87 (32.1)	96 (55.2) *
Non-adherence or misuse of inhaler	152 (34.2)	117 (43.2)	35 (20.1) *
Exposure of allergens/air pollution	77 (17.3)	42 (15.5)	35 (20.1)
Comorbidity	57 (12.8)	33 (12.2)	24 (13.8)

Data are presented with numbers of responders (% of each column). * *p* < 0.05, chi-square, vs. non-pulmonologists.

**Table 3 biomedicines-10-03253-t003:** Differential responses between pulmonologists and non-pulmonologists.

Questions	Cut-Off Value	Non-Pulmonologist (*n* = 271)	Pulmonologist (*n* = 174)	*p* *
**Section 1. Oral corticosteroid (OCS) for acute exacerbation (AE)**				
How many percent of your asthma patients need OCS to treat AEs per month?	≥6 %	153 (56.5)	94 (54)	0.342
What daily dose of OCS (prednisolone or equivalent in mg) do you prescribe to treat an AE?	≥20 mg	121 (44.6)	102 (58.6)	0.005
How many days of OCS do you prescribe to treat an AE?	≥3 days	162 (59.8)	159 (91.4)	<0.001
**Section 2. Regular corticosteroid (OCS) for asthma control**				
How many percent of your patients need regular OCS for asthma control per month?	≥6 %	79 (29.2)	58 (33.3)	0.4
What daily dose of OCS (prednisolone or equivalent in mg) do you prescribe as regular doses for asthma control?	≥6 mg	147 (54.2)	117 (67.2)	0.008
How many months do each treatment course usually take while you prescribe regular OCS for asthma control per year?	≥1 month	80 (29.5)	80 (46)	0.001
How many percent of your asthma patients experience ≥2 courses of regular OCS for asthma control per year?	≥6 %	133 (49.1)	97 (55.7)	0.175
**Section 3: Asthma control status**				
How many percent of your asthma patients had ≥2 unscheduled visits to clinic or emergent department due to AEs in the past year?	≥1 %	221 (81.5)	163 (93.7)	<0.001
How many percent of your asthma patients were intubated or admitted to intensive care units due to AEs in the past year?	≥1 %	76 (28)	106 (61)	<0.001
How many percent of your asthma patients used ≥3 canisters of short-acting β2 agonists (SABA) in the past year?	≥6 %	113 (41.7)	109 (62.6)	<0.001
How many percent of your asthma patients may benefit from biological agents based on reductions of either regular doses of OCS or frequencies of AE?	≥6 %	55 (20.3)	60 (34.5)	<0.001

Data are presented with numbers of responders (% of each column). * Chi-square.

## Data Availability

Data available on request due to restrictions of privacy.

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
