# Peer review of "Unmet Need for Oral Corticosteroids Use and Exacerbations of Asthma in Primary Care in Taiwan"

_biomedicines, 2022, doi:10.3390/biomedicines10123253_

Round 1

Reviewer 1 Report

The manuscript "Unmet need for oral corticosteroids use and exacerbations of 2 asthma in primary care in Taiwaw" By Chen et al.,

provides the results of a cross-sectional survey to invited physicians that were board-certified to treat asthma in adult patients and practised in primary health care.  They evaluated the prescription of OCS to treat acute exacerbations and acute exacerbations adverse outcomes, and use of OCS for asthma control. It was concluded that, although there is a benefit for certain population by the use of biologicals, patients consider OCS as the first choice of asthma treatment. Reconsideration of methods to treat asthma should be applied in Taiwan.

The study is well presented, with the introduction to support the scope of the study and the discussion to link appropriately earlier with knowledge brought up by the current investigation.

Author Response

Dear  Reviewer,

Thank you very much indeed for your valuable comments!

Reviewer 2 Report

This is an interesting manuscript on the real world treatment of asthma in Taiwan. My comments regard further clarification of the  study.

Abstract, line 23 - please give the number of physicians who were contacted.  This is important because the information will help to understand how representative the responses are, e.g. 10% answered, 50%, 80%...

Line 24-35: instead of high, low proportion of patients, please indicate percentage.

Figure 1B, 2B, question 2 of table 3: The caption asks for the number of OCS doses per day - like once daily, twice daily.  I think that the figure indicates the the daily dose of OCS - like What is the daily dose of OCS (mg prednisone or equivalent).

Table 3: the Column 'response' is hard to understand, is this the summary of all answers ?

Minor:

Line 46 to

Line 233 figure legend - the word is misplaced

Author Response

Responses to Reviewer 2 Comments

Manuscript no. biomedicines-2026648

Date: Nov. 27 2022

Dear  Reviewer,

Thank you very much indeed for your valuable comments and suggestions. The following is to address concerns from Reviewers. We hope this revised manuscript will meet the standard of acceptance for publication.

Point 1: Abstract, line 23 - please give the number of physicians who were contacted.  This is important because the information will help to understand how representative the responses are, e.g. 10% answered, 50%, 80%...

Response 1: Thank you for your comments. There were 445 out of 500 physicians who responded to our questionnaire. We have revised the manuscript (Page 1 in red)

Point 2: Line 24-35: instead of high, low proportion of patients, please indicate percentage.

Response2: Thank you for your comments. After recalculation, we have revised the whole section in the manuscript (Page 1, 2 in red).

Point 3: Figure 1B, 2B, question 2 of table 3: The caption asks for the number of OCS doses per day - like once daily, twice daily.  I think that the figure indicates the daily dose of OCS - like What is the daily dose of OCS (mg prednisone or equivalent).

Response 3: Thank you for your comments. We have revised the captions of Figure 1B, 2B, and Question 2 of Table 3 in the manuscript. (Page 6, 8 and 11 in red)

Point 4: Table 3: the Column 'response' is hard to understand, is this the summary of all answers?

Response 4: Thank you for your comments. We have replaced “response” with “cut-off value” in the caption and added the units in the “cut-off value” column of Table 3. Please refer to the Page of the revised manuscript. (Page 11 in red)

Point 5: Line 46 to

Response 5: Thank you for your comments. We have corrected the typo and the new text reads 4% to 65%. Please refer to the Page of the revised manuscript. (Page 2 in red)

Point 6: Line 233 figure legend - the word is misplaced

Response 6: Thank you for your comments. We have deleted the typo “figure legend”. (Page 17)